# Direct Photocoagulation for Treating Microaneurysms with Hyperreflective Ring in Eyes with Refractory Macular Edema Associated with Branch Retinal Vein Occlusion

**DOI:** 10.3390/jcm11030823

**Published:** 2022-02-03

**Authors:** Hirofumi Sasajima, Masahiro Zako, Yoshiki Ueta, Hideo Tate, Chisato Otaki, Kenta Murotani, Takafumi Suzuki, Hidetoshi Ishida, Yoshihiro Hashimoto, Naoko Tachi

**Affiliations:** 1Department of Ophthalmology, Shinseikai Toyama Hospital, Imizu 939-0243, Japan; ueta@shinseikai.or.jp (Y.U.); tate@shinseikai.or.jp (H.T.); c-ootaki@shinseikai.or.jp (C.O.); etoile1914100@gmail.com (T.S.); ishi1214@kanazawa-med.ac.jp (H.I.); hashimoto@shinseikai.or.jp (Y.H.); ntachihana@gmail.com (N.T.); 2Department of Ophthalmology, Asai Hospital, Seto 489-0866, Japan; mzako@aol.com; 3Biostatistics Center, Kurume University, Kurume 830-0011, Japan; kmurotani@med.kurume-u.ac.jp; 4Department of Ophthalmology, University of Tokyo Hospital, Bunkyo-ku, Tokyo 113-8655, Japan; 5Department of Ophthalmology, Kanazawa Medical University, Kahoku 920-0293, Japan

**Keywords:** branch retinal vein occlusion, refractory macular edema, hyperreflective ring, microaneurysms, direct photocoagulation, optical coherence tomography

## Abstract

Microaneurysms (MAs) with hyperreflective rings are sometimes detected in eyes with refractory macular edema (ME) associated with branch retinal vein occlusion (BRVO) for more than 12 months after onset when examined using optical coherence tomography (OCT). We proposed that these MAs could result in refractory ME secondary to BRVO and hypothesized that OCT-guided direct photocoagulation of MAs could result in a reduction in refractory ME. Eleven eyes (from eleven different patients) with refractory ME associated with BRVO for more than 12 months following initial treatment were included. The mean number of MAs in each eye at baseline was 3.5 ± 2.0 (range, 1–8). The mean central subfield thickness, central macular volume, and parafoveal macular volume significantly decreased 6 months following initial direct photocoagulation when compared with those at baseline (baseline = 378.7 ± 61.8 μm, post-treatment = 304.2 ± 66.7 μm, *p* = 0.0005; baseline = 0.3 ± 0.049 mm^3^, post-treatment = 0.24 ± 0.053 mm^3^, *p* = 0.001; and baseline = 2.5 ± 0.14 mm^3^, post-treatment = 2.28 ± 0.15 mm^3^, *p* = 0.001, respectively). Moreover, the mean best-corrected visual acuity significantly improved 6 months following initial direct photocoagulation when compared with that at baseline (baseline = 0.096 ± 0.2 logarithm of the minimum angle of resolution (logMAR), post-treatment = 0.0077 ± 0.14 logMAR, *p* = 0.031). Direct photocoagulation could be suggested as a treatment option for refractory ME associated with BRVO in MAs with a hyperreflective ring on OCT.

## 1. Introduction

Macular edema (ME) is the leading cause of vision loss associated with branch retinal vein occlusion (BRVO) [1,2,3]. The efficacy of anti-vascular endothelial growth factor (anti-VEGF) drugs for ME secondary to BRVO has been reported [4,5,6]; however, despite patients receiving multiple intravitreal injections of anti-VEGF drugs, ME persists for over 1 year following treatment [7,8]. Retinal vascular abnormalities, such as microaneurysms (MAs) [9,10], can develop in the chronic phase of BRVO. MAs can cause refractory MEs in BRVO [9,10]. Although anti-VEGF therapy is considered the first line of treatment for ME in the acute phase of BRVO [4,5,6,7,8], no consensus exists on the treatment for refractory ME.

Sakimoto et al. [10] reported that fluorescein angiography (FA)-guided direct photocoagulation of MAs is effective in treating chronic ME in patients with BRVO. Additionally, recently, focal/grid laser photocoagulation has been performed without FA guidance [11,12] since the leakage area on FA and retinal thickening on optical coherence tomography (OCT) did not completely coincide occasionally [13].

MAs can be visible on OCT images as capsular structures (ring signs) in diabetic retinopathy [14,15,16,17,18,19]. In eyes with refractory ME associated with BRVO, we sometimes detected MAs with hyperreflective rings on OCT images similar to those in diabetic retinopathy. We reviewed the OCT images of eyes with chronic ME associated with BRVO in our hospital and found that some eyes had MAs with hyperreflective rings within the perimeter of the ME on the OCT images. Shin et al. [18] reported that OCT-guided selective focal laser photocoagulation for MAs with diabetic ME showed similar anatomic and functional outcomes compared with conventional laser treatment, with significantly less retinal damage. However, the efficacy of OCT-guided direct photocoagulation for MAs in BRVO is not well characterized, and further evaluation is necessary.

In this study, we elucidated the efficacy of direct photocoagulation for MAs with hyperreflective rings on OCT images in eyes with refractory ME secondary to BRVO, without FA guidance.

## 2. Materials and Methods

### 2.1. Patients and Examinations

This retrospective case series study included patients with refractory ME associated with BRVO for more than 12 months following initial treatment. Each patient received oral and written information about the laser treatment and provided consent prior to treatment. The Institutional Review Board of Shinseikai Toyama Hospital approved the study protocol (reference number: 211201-1), which adhered to the tenets of the Declaration of Helsinki. We reviewed the medical and ocular histories of patients with refractory ME owing to BRVO treated with OCT-guided direct photocoagulation between 9 April 2021 and 26 November 2021 in the Department of Ophthalmology of the Shinseikai Toyama Hospital (Toyama, Japan).

The inclusion criteria were as follows: patients who had a hyperreflective ring on spectral-domain OCT (Spectralis OCT, Heidelberg Engineering, Heidelberg, Germany) images with ME owing to BRVO, had ME for more than 12 months following initial treatment, and could be followed for 6 months following initial direct photocoagulation. The exclusion criteria included the following: patients with the presence of hemicentral retinal vein occlusion, significant media opacity, and other retinal disorders such as diabetic retinopathy. Patients who had undergone ocular surgery, such as cataract surgery, vitrectomy, and grid or scatter photocoagulation, within 6 months and during the current study; had undergone treatment, including an intravitreal drug injection such as ranibizumab (Lucentis^®^, Genentech Inc., South San Francisco, CA, USA), aflibercept (Eylea^®^, Regeneron Pharmaceuticals, Inc., Tarrytown, NY, USA), and triamcinolone acetonide (TA) (MaQaid^®^, Wakamoto Pharmaceutical Co., Ltd., Tokyo, Japan); or were treated with a sub-Tenon injection of TA (STTA) (Kenacort^®^, Bristol-Myers Squibb, Tokyo, Japan) at least 3 months following the treatment were also excluded.

During the study period, all of the patients underwent comprehensive ophthalmic examinations, including best-corrected visual acuity (BCVA) measurement using a Landolt C-chart, slit-lamp biomicroscopy, and indirect ophthalmoscopy. OCT was used to evaluate the central subfield thickness (CST) and macular volume before and at 3 and 6 months following direct photocoagulation. The number of baseline MAs with hyperreflective rings within the ME perimeter was counted on 19-line raster scans centered on the fovea. Similar to a previous study [19], we measured the outer diameter of the MA, including the wall, by placing the calipers of the built-in measurement software of the Spectralis parallel to the optical axis. The distance between the MA closest to the fovea was measured. Forty-five-degree color fundus photographs were obtained using a fundus camera (TRC-NW8, Topcon, Tokyo, Japan) before and after direct photocoagulation.

### 2.2. Assessment of the Hyperreflective Rings and Laser Setting

We obtained 19-line sequential raster scans centered on the fovea to evaluate the presence of MAs with hyperreflective rings on the OCT images (Figure 1). If we detected an ME outside the raster scans, we used OCT B-scan images centered on the ME to detect the hyperreflective rings. Direct photocoagulation was not performed on MAs detected outside the perimeter of the ME. Where we detected MAs with hyperreflective rings within the ME perimeter, the locations of MAs on the OCT images precisely corresponded with their locations on the color fundus images (Figure 2). We then performed direct photocoagulation for MAs using a 577 nm laser (EasyRet^®^, Quantel Medical, Cournon-d’Auvergne, France) with the following parameters: spot size of 50–100 μm (smaller than the size of MA to avoid damage to surrounding tissue), laser power set to 90–150 mW (so that only the MA became gray or white), and pulse duration between 10 and 20 ms using a direct contact laser lens (Centralis Direct^®^, Volk, Mentor, OH, USA). In general, the typical funduscopic aspect of MAs was a minute reddish spot surrounded by a faint whitish wall (Figure 2B) [19]. Direct photocoagulation of the MAs was performed when the CST exceeded 250 μm. All direct photocoagulations were performed by two retinal specialists (H. S. performed the laser treatment in cases 1, 5–8, 10, and 11; Y. U. performed laser treatment in cases 2–4, 9). When ME persisted and OCT images showed remnants or newly developed hyperreflective rings following direct photocoagulation, additional laser treatment was performed by two surgeons when deemed necessary.

### 2.3. Assessment of Central Macular Volume and Parafoveal Macular Volume by Optical Coherence Tomography Map

We analyzed the macular volume using the standardized Early Treatment Diabetic Retinopathy Study grid on the OCT map in addition to the CST to evaluate the efficacy of direct photocoagulation. Central macular volume (CMV) and parafoveal macular volume (PFMV) were defined as the volume of the central fovea (circle with a 1 mm diameter) and the volume of the parafovea (ring between 1 and 3 mm surrounding the fovea), respectively.

### 2.4. Endpoints

The primary endpoint was the change in the mean CST 6 months following the initial laser treatment. Secondary endpoints were the changes in the CMV, PFMV, and BCVA 6 months following the initial laser treatment. Complications arising from the laser treatment were also assessed.

### 2.5. Statistical Analysis

A biostatistician (K.M.) performed the statistical analyses using SAS version 9.4 software (SAS Institute, Inc., Cary, NC, USA). The normality was evaluated by performing a Shapiro–Wilk test. After confirming that the data were approximately normally distributed, all of the values are expressed as mean ± standard deviation. Decimal BCVA was converted to the logarithm of the minimal angle of resolution (logMAR) units for analysis. The Wilcoxon signed-rank test was used to compare the baseline BCVA, CST, CMV, and PFMV values before and 3 and 6 months following the initial laser treatment. We also performed a post hoc power analysis for each endpoint. Spearman’s correlation coefficients were calculated to determine the significance of the association between the largest MA size and the effect of direct photocoagulation. *p* < 0.05 was considered statistically significant.

## 3. Results

### 3.1. Patient Characteristics

Eleven eyes from eleven different patients met the study criteria for analysis. A summary of the patient characteristics and measurements made using OCT is presented in Table 1. The characteristics of individual patients, their clinical history, and the treatment they received are summarized in Table 2. In the 6 months following initial direct photocoagulation, additional direct photocoagulation was performed in five eyes (45.5%) owing to confirmation of remnants or newly developed hyperreflective rings on the OCT images. The mean number of additional laser treatments per patients was 0.91 ± 1.1 (Table 2).

### 3.2. Changes in the Central Subfield Thickness, Central Macular Volume, and Parafoveal Macular Volume

Figure 3 shows images of all cases included in this study at baseline (A–D) and 6 months following initial direct photocoagulation (E–H). The mean CST at 3 and 6 months (316.1 ± 64.5 μm and 304.2 ± 66.7 μm) following initial laser treatment significantly (*p* = 0.001 and *p* = 0.0005) decreased from the baseline (378.7 ± 61.8 μm) (Figure 4). The mean CMV at 3 and 6 months (0.26 ± 0.052 mm^3^ and 0.24 ± 0.053 mm^3^) following initial laser treatment significantly (*p* = 0.0049 and *p* = 0.001) decreased from the baseline (0.3 ± 0.049 mm^3^) (Figure 4). The mean PFMVs at 3 and 6 months (2.3 ± 0.13 mm^3^ and 2.3 ± 0.15 mm^3^) following initial laser treatment also significantly (*p* = 0.001 and *p* = 0.001) decreased from the baseline (2.5 ± 0.14 mm^3^) (Figure 4). The post hoc powers were greater than 0.999 for CST, CMV, and PFMV in the comparison between that at baseline and 6 months following initial laser treatment.

### 3.3. Correlation of the Largest Microaneurysm Size with Central Subfield Thickness, Central Macular Volume, and Parafoveal Macular Volume

MA size was not significantly correlated with the baseline CST (ρ = −0.101, *p* = 0.77), CMV (ρ = −0.023, *p* = 0.95), or PFMV (ρ = 0.329, *p* = 0.32) or with the reduction in magnitude of the change in the CST (ρ = 0.337, *p* = 0.31), CMV (ρ = 0.37, *p* = 0.26), or PFMV (ρ = 0.247, *p* = 0.46) at 6 months following initial direct photocoagulation.

### 3.4. Changes in the Best-Corrected Visual Acuity and Complications

The mean BCVA at 3 months (0.028 ± 0.16 logMAR) following the initial laser treatment did not significantly (*p* = 0.16) improve from the baseline (0.096 ± 0.2 logMAR) (Figure 4). However, the mean BCVA at 6 months (0.0077 ± 0.14 logMAR) following the initial laser treatment showed significant (*p* = 0.031, post hoc power = 0.712) improvement (Figure 4). In this study, no complications occurred during laser treatment in any patient.

## 4. Discussion

In this study, we demonstrated that OCT-guided direct photocoagulation for MAs was effective in eyes with refractory ME associated with BRVO. At 3 months following initial direct photocoagulation for MAs with a hyperreflective ring on the OCT images, the CST, CMV, and PFMV significantly decreased from the baseline, and the effect was maintained for at least 6 months following the procedure. Moreover, BCVA at 6 months following the initial direct photocoagulation significantly improved from baseline. Our results suggest that MAs with hyperreflective rings within the perimeter of the ME could cause refractory ME supporting the adoption of direct photocoagulation for the treatment of refractory ME associated with BRVO.

Previous studies have reported that direct photocoagulation is effective in treating MAs in eyes with refractory ME associated with BRVO [9,10]. However, these studies did not mention whether MAs with a hyperreflective ring on OCT images cause MEs. In this study, we hypothesized that MAs with a hyperreflective ring on OCT images within the ME perimeter cause refractory ME owing to BRVO. By treating the MA using direct photocoagulation, we were able to treat the refractory ME owing to BRVO, thereby demonstrating a causal relationship between MA and ME due to BRVO.

Meanwhile, it is necessary to consider the difference between the efficacy of OCT-guided laser and conventional FA-guided laser. Table 3 shows a comparison between this study and the most relevant previous study [10] in which BRVO cases with ME persisting for more than one year were treated with FA-guided direct photocoagulation. As shown in Table 3, it is necessary to take into consideration the difference in the time for the initial direct photocoagulation and the follow-up period between the current study and the previous study [10]; moreover, in the previous study [10], direct photocoagulation was performed not only for highly permeable MAs but also for dilated and leaking capillaries. However, we found that the CST and BCVA improved in a similar fashion following the initial laser treatment. These results suggest that the OCT-guided direct photocoagulation for MAs is as effective as conventional FA-guided direct photocoagulation.

The reasons for the effectiveness of direct photocoagulation for MAs with a hyperreflective ring should be evaluated. Previous studies have suggested that the high intensity of the ring is attributed to the subclinical features of lipoprotein extravasation following the breakdown of the inner blood–retinal barrier [14] and composition of cellular components [15], such as erythrocytes, leukocytes, and lipid deposits, in the hyperreflective ring in the MA lumen when examined using electron microscopy [20]. Higher concentrations of hemoglobin-containing cellular components increase laser absorption [21,22] and may subsequently improve MA resolution. Increased intraluminal reflectance following direct photocoagulation suggests photothrombosis [19]. Therefore, direct photocoagulation of MAs with a hyperreflective ring may be effective in stopping the leakage source of refractory ME associated with BRVO.

Horii et al. [15] classified MAs into three types according to the status of the capsular structure (i.e., the ring is complete, incomplete, or absent) and reported that MAs with a complete ring sign were accompanied by nearby cystoid spaces less frequently than when an incomplete or no ring sign was present. This report implied that differences in the type of MAs with a hyperreflective ring could influence the outcome of direct photocoagulation. Further studies are needed to elucidate whether the MA type is related to the outcome of direct photocoagulation.

The size of MA might affect the outcome of direct photocoagulation as well. The largest MA size was measured in each case. The average MA was relatively large (mean size, 134.3 ± 51.5 μm). It is relatively easy to perform direct photocoagulation on large MAs since they are easily detected on color fundus images. Meanwhile, the effect of direct photocoagulation on MAs smaller than 130 μm was limited as previously reported [19]. However, in this study, the largest MA size did not correlate with the baseline CST, CMV, and PFMV or with the reduction in the magnitude of the change in the CST, CMV, or PFMV at 6 months following initial direct photocoagulation. This may have been due to other MAs with hyperreflective rings within the ME. Retinal thickness may depend on vascular permeability, retinal ischemia, or cytoplasmic swelling of Müller cells. Further large-scale studies are warranted to elucidate the relationship between MA size and the effect of direct photocoagulation.

Patients included in this study were treated with anti-VEGF drugs and/or TA before initial laser treatment. The number of MAs was many in eyes treated with TA, as previously reported [9]. Prospective studies are warranted to determine whether drug differences affect MA formation in BRVO.

No complications were observed during the laser treatment in this study. The distance between the fovea and the closest MA with a hyperreflective ring was relatively long (mean distance, 1966.7 ± 738.2 μm). However, when performing direct photocoagulation on MAs close to the fovea and/or in patients with poor fixation, the risk of foveal burns should be considered. OCT-guided direct photocoagulation can help determine the precise locations of targeted MAs and may reduce this risk.

This study had several limitations. First, owing to the retrospective study design and small sample size, a sampling bias was present. Although the number of cases in this study was limited, we were able to confirm that the powers were statistically sufficient. Second, although no other treatment, such as intravitreal injections of anti-VEGF drugs, was needed during the 6 months follow-up period, a longer follow-up period could validate the longevity of the effect of direct photocoagulation. Third, in this study, we included patients with refractory ME and those with MAs with hyperreflective rings within the ME on OCT images in eyes with BRVO for more than 12 months following initial treatment. A previous study reported [9] that MAs were detected in 51% of the patients within 6 months of BRVO onset and in 84% of the patients within 9 months. However, to date, the appropriate timing for effective direct photocoagulation for MAs has not been determined in patients with BRVO. Our study suggests that the appropriate timing for direct photocoagulation might be suitable when MAs with hyperreflective rings can be detected on the OCT images. Finally, in this study, we did not compare OCT-guided direct photocoagulation with conventional FA-guided direct photocoagulation. However, OCT-guided direct photocoagulation in this study was considered as effective as previously reported [10] conventional FA-guided direct photocoagulation. Thus, our preliminary yet promising results could encourage comparative and/or combination studies to investigate the effect of OCT-guided direct photocoagulation on refractory ME associated with BRVO.

Despite these limitations, our study had several strengths. MAs with hyperreflective rings can be easily detected on OCT B-scan images, and the OCT-guided direct photocoagulation for MAs can be performed even in patients where we cannot examine FA or perform intravitreal injection of anti-VEGF drugs owing to the high risk of cerebrovascular or cardiovascular complications. In addition, direct photocoagulation for MAs could be a definitive treatment for refractory ME in BRVO, unlike symptomatic treatments such as intravitreal injections of anti-VEGF drugs and TA.

In conclusion, direct photocoagulation can aid in the treatment of MAs with hyperreflective rings on OCT images and can be considered an effective treatment option for refractory ME associated with BRVO.

## Figures and Tables

**Figure 1 jcm-11-00823-f001:**
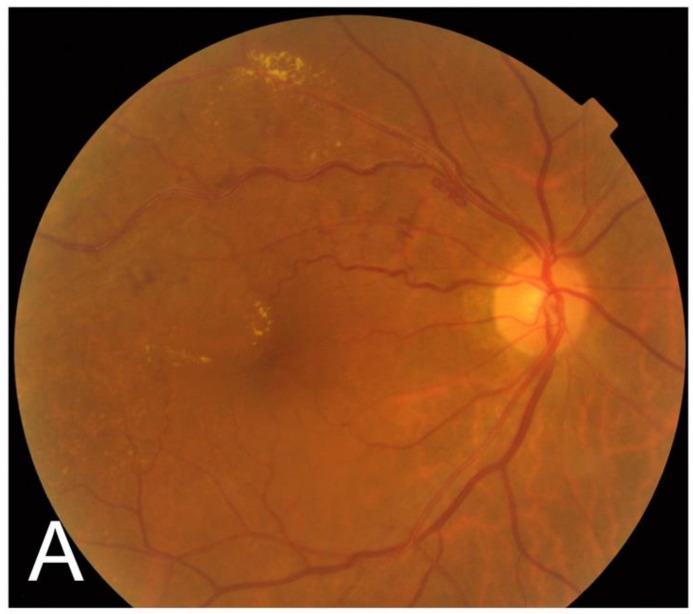
Investigating all of the hyperreflective rings (microaneurysm) on optical coherence tomography (OCT) images using a raster scan (case 9). Color fundus image (**A**) and OCT map (**B**) were examined before initial direct photocoagulation. The horizontal green line in the OCT map (**B**) corresponds to the OCT B-scan image (**C**). A hyperreflective ring (arrow) was seen as a capsular structure in the OCT B-scan image (**C**).

**Figure 2 jcm-11-00823-f002:**
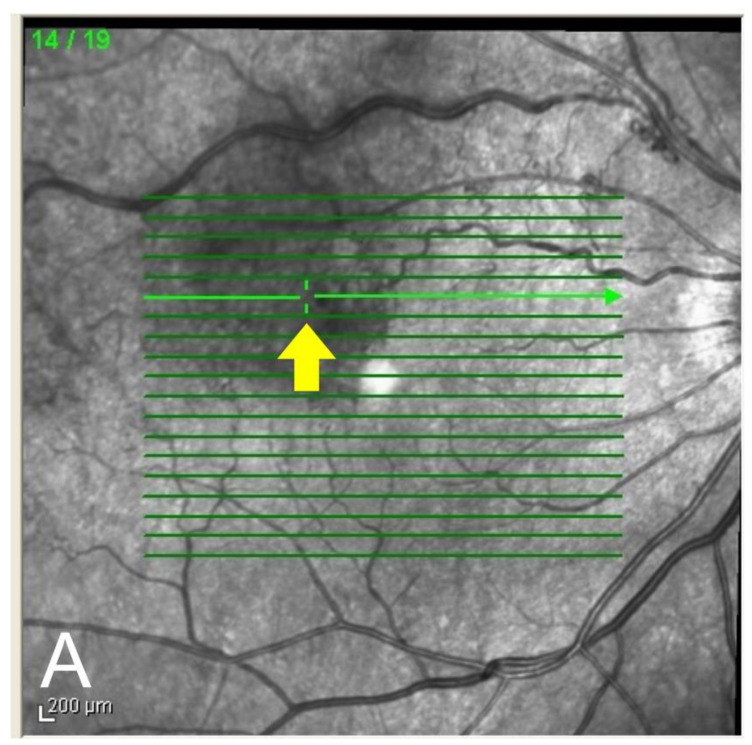
Superimposition of a microaneurysm (MA) on an optical coherence tomography B-scan image corresponding to the color fundus image (case 9). Green slider (arrow) on the thicker, horizontal green line in the infrared image (**A**) corresponds to the location of the color fundus image (arrowhead) (**B**). MA looked similar to a minute reddish spot surrounded by a faint whitish wall on the color fundus image (**B**).

**Figure 3 jcm-11-00823-f003:**
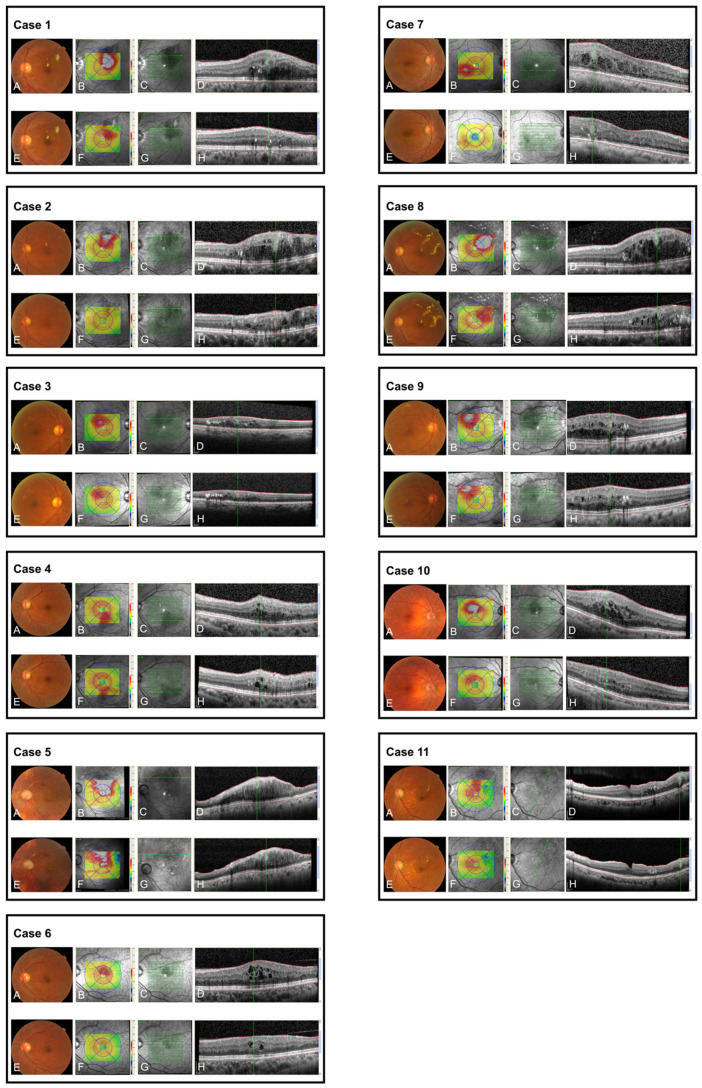
Images of all cases included in this study at baseline (**A**–**D**) and 6 months following initial direct photocoagulation (**E**–**H**). (**A**) Color fundus image before the initial direct photocoagulation. (**B**) Optical coherence tomography (OCT) map showing the macular edema region. (**C**) The 19-line raster scans centered on the fovea in the infrared image (except for cases 5 and 11). The thicker, green horizontal line and green slider on the green line in the infrared image in (**C**) correspond to the location of the green line and the green slider in the OCT map in (**B**) (except for cases 5 and 11). (**D**) OCT-B scan images. The OCT-B scan image was taken at the plane of the thicker green horizontal line in (**C**), and the vertical, green line in the OCT B-scan corresponds to the location of the green slider on the thicker green line in the infrared image in (**C**). A hyperreflective ring is observed on the green line in the OCT B-scan image. (**E**) Color fundus image obtained 6 months following initial direct photocoagulation. (**F**) The location of the thicker horizontal green line and green slider on the horizontal green line on the OCT map corresponds to the location of the green line and the green slider in the OCT map in (**B**) (except for cases 5 and 11). (**G**) The green slider on the green line in the infrared image corresponds to the location in (**C**). (**H**) The OCT-B scan image corresponds to the plane of the green line in (**G**) and the vertical, green line in the OCT image corresponds to the green slider in (**G**).

**Figure 4 jcm-11-00823-f004:**
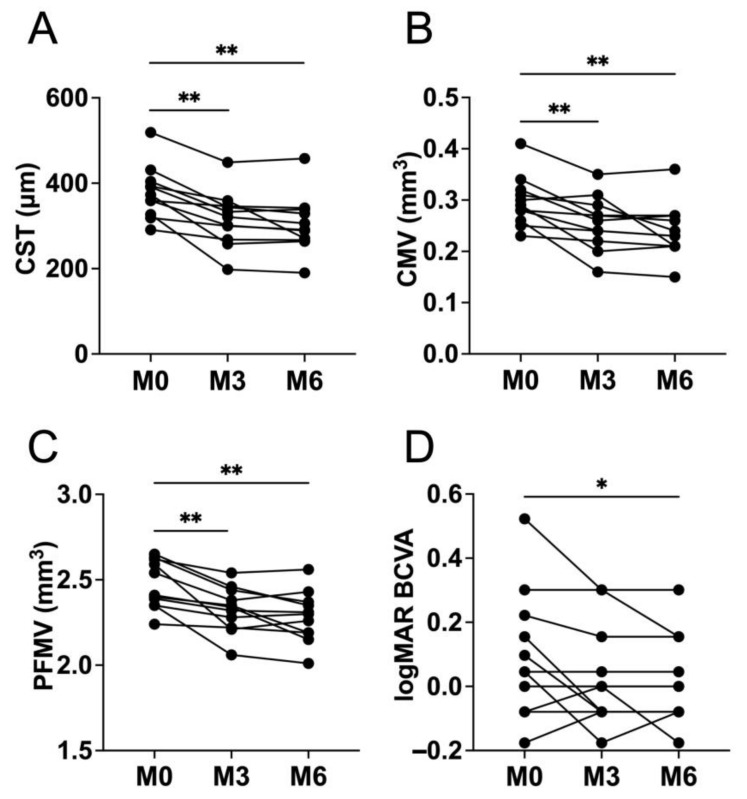
Changes in the central subfield thickness (CST), central macular volume (CMV), parafoveal macular volume (PFMV), and best-corrected visual acuity (BCVA). The mean CST (**A**), CMV (**B**), and PFMV (**C**) at 3 and 6 months following initial direct photocoagulation significantly decreased from the baseline. The mean logarithm of the minimum angle of resolution (logMAR) BCVA (**D**) at 3 months following initial direct photocoagulation did not significantly improve from the baseline. However, at 6 months following initial direct photocoagulation, we observed a significant improvement compared with baseline. M0, baseline; M3, month 3; M6, month 6. * *p* < 0.05. ** *p* < 0.01.

**Table 1 jcm-11-00823-t001:** Summary of patient characteristics and optical coherence tomography findings before initial direct photocoagulation.

Parameter	Value
No. of eyes	11
Age (years)	70 ± 12.8
Sex (male/female)	3/8
Eye (right/left)	4/7
Lens status (phakic/pseudophakic)	6/5
Duration before initial laser treatment (months)	42 ± 35.3
logMAR BCVA	0.096 ± 0.2
Central subfield thickness (μm)	378.7 ± 61.8
Central macular volume (mm^3^)	0.3 ± 0.049
Parafoveal macular volume (mm^3^)	2.5 ± 0.14
No. of hyperreflective rings	3.5 ± 2.0
Largest MA size (μm)	134.3 ± 51.5
Distance between the fovea and closest MA (μm)	1966.7 ± 738.2

Data are expressed as the mean ± standard deviation. BCVA, best-corrected visual acuity; logMAR, logarithm of the minimum angle of resolution; No., number; MA, microaneurysm.

**Table 2 jcm-11-00823-t002:** Characteristics of individual patients undergoing direct photocoagulation for microaneurysm (MA) and clinical course following direct photocoagulation.

Case	Age	Period (M)	Previous Treatment (Times)	No. of MA	Largest MA (μm)	Distance to Fovea (μm)	logMAR BCVA			CST (μm)			CMV (mm^3^)			PFMV (mm^3^)			Additional Laser
							0 M	3 M	6 M	0 M	3 M	6 M	0 M	3 M	6 M	0 M	3 M	6 M	
1	63	12	Anti-VEGF (3), STTA (1)	5	85	1861	−0.079	0	−0.18	431	344	329	0.34	0.27	0.26	2.65	2.46	2.35	3 M
2	71	94	Anti-VEGF (40), scatter laser	3	205	2193	0.046	0.046	0.046	391	321	306	0.31	0.29	0.24	2.41	2.34	2.15	None
3	58	30	Anti-VEGF (10)	4	110	980	0	0	0	404	333	339	0.32	0.26	0.27	2.54	2.38	2.43	3, 4 M
4	78	24	Anti-VEGF (9), scatter laser	1	88	1769	−0.18	−0.079	−0.079	291	268	267	0.23	0.22	0.21	2.39	2.32	2.31	None
5	88	51	Anti-VEGF (17), scatter laser	4	199	2684	0.52	0.3	0.3	519	449	458	0.41	0.35	0.36	2.62	2.54	2.56	1, 3, 4 M
6	58	12	Anti-VEGF (3), scatter laser	2	85	960	0.15	−0.079	−0.079	393	359	271	0.3	0.31	0.21	2.4	2.35	2.19	None
7	71	14	Anti-VEGF (3)	2	143	1887	0.22	0.15	0.15	327	198	190	0.26	0.16	0.15	2.35	2.06	2.01	None
8	48	33	Anti-VEGF (3), IVTA (6), scatter laser	8	140	2036	0.097	−0.079	−0.079	359	301	289	0.28	0.24	0.23	2.63	2.44	2.37	1, 3 M
9	71	113	Anti-VEGF (16)	3	113	2415	0.046	−0.18	−0.079	319	300	291	0.25	0.24	0.23	2.35	2.28	2.3	1, 3 M
10	90	12	Anti-VEGF (3)	5	90	1372	0.3	0.3	0.15	373	258	264	0.29	0.2	0.21	2.59	2.21	2.26	None
11	74	67	Anti-VEGF (3), vitrectomy, scatter laser	2	219	3477	−0.079	−0.079	−0.079	359	346	342	0.28	0.27	0.27	2.24	2.22	2.19	None

The period was between the initial visit and the initial laser treatment. Scatter laser photocoagulation was performed to treat an area of nonperfusion in six eyes. BCVA, best-corrected visual acuity; logMAR, logarithm of the minimum angle of resolution; CST, central subfield thickness; CMV, central macular volume; PFMV, parafoveal macular volume; M, month; No., number; VEGF, vascular endothelial growth factor; STTA, sub-tenon injection of triamcinolone acetonide; IVTA, intravitreal injection of triamcinolone acetonide.

**Table 3 jcm-11-00823-t003:** Comparison of the treatment effects of optical coherence tomography (OCT)-guided and fluorescein angiography (FA)-guided direct photocoagulation for microaneurysms.

	OCT-Guided Laser (This Study)	FA-Guided Laser *
No. of eyes	11	16
Age (years)	70 ± 12.8	72.0 ± 6.3
Period between the initial visit and the initial laser treatment (M)	42 ± 35	20.9 ± 9.9
Follow-up period from the initial laser treatment (M)	6	20.3 ± 8.0
No. of applications of laser treatment	1.9 ± 1.1	1.7 ± 0.9
Central subfield thickness (μm)		
Baseline	378.7 ± 61.8	465.0 ± 107.6
3 M following initial laser treatment	316.1 ± 64.5	355.3 ± 91.3
6 M following initial laser treatment	304.2 ± 66.7	334.3 ± 68.8
BCVA (logMAR)		
Baseline	0.096 ± 0.2	0.39 ± 0.28
3 M following initial laser treatment	0.028 ± 0.16	0.31 ± 0.23
6 M following initial laser treatment	0.0077 ± 0.14	0.24 ± 0.22

* The results of FA-guided laser for microaneurysms were quoted from a previous report [10] for comparison with our results. Data are expressed as the mean ± standard deviation. BCVA, best-corrected visual acuity; logMAR, logarithm of the minimum angle of resolution; M, month; No., number.

## Data Availability

The datasets used in this study are available from the corresponding author upon reasonable request.

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
