# Peer review of "Direct Photocoagulation for Treating Microaneurysms with Hyperreflective Ring in Eyes with Refractory Macular Edema Associated with Branch Retinal Vein Occlusion"

_jcm, 2022, doi:10.3390/jcm11030823_

Round 1

Reviewer 1 Report

I am very grateful to be able to read and review such a great work.   

The authors described an interesting, comprehensive and informative work focused on direct photocoagulation for treating microaneurysms. Authors cover a very important and clinically significant part of everyday life in the retina service.  

Clear and focused research questions have been stated in a proper and easy to revise manner. The text is concise and can be read with ease. 

There is a need for some major and some minor revision: 

  1. This is rather case series than full article and should be stated as such. I am aware that this is a difficult to gather subject for a coherent research sample, but the sample size analysis should be performed as the previous research data are available and can be used for this purpose.  
  1. Figure 1 – there are too much data the double divided figure is confusing please divide it into 2 separate figures. Additionally, the good quality fundus photography with full ring properties of the MA should be indicated in it. 
  1. I am a bit concerned about the statistical presentation. The mean and SD for the 11 patients included? If the normality of the data can be proven (e.g., with Shapiro-Wilk test) this is an option but no information in the statistical analysis mention that part. However, if the normality cannot be proven median and 95%CI is better option possibly with mode – you used the non-parametric test. 
  1. The authors mention the previous studies in the discussion (line 230 – 236) but no exact comparison can be performed. Please provide the extract of the most relevant study preferably in table or graph for the reader to be able to compare the present study with previous ones and to argumentation of the discussion conducted in the text.  
  1. Creation of the figure (graphical equivalent that can be used as graphical abstract) describing the possible mechanisms of creation of MAs and their impact on chronic ME is encouraged, preferably with the changes introduced by laser treatment, but not obligatory. This could greatly improve the visibility of the work.  
  1. In the one of the last parts of the discussion authors indicate the limitation of their study. Please, provide the defence for each limitation – some are given but not quite sufficient. 

Author Response

Response to Reviewer 1 Comments

Point 1: This is rather case series than full article and should be stated as such. I am aware that this is a difficult to gather subject for a coherent research sample, but the sample size analysis should be performed as the previous research data are available and can be used for this purpose.

Response 1: Thank you for pointing this out. We completely agree with your comment. As per your suggestions, we have added the term “case series” to the revised manuscript (Page 2, line 21). We have also performed a post-hoc power analysis for visual acuity (VA), central subfield thickness (CST), central macular volume (CMV), and parafoveal macular volume (PFMV). We have added this information to the revised manuscript (Page 6, lines 19–20). The post-hoc power was 0.712 for logMAR VA in the comparison between that at baseline and 6 months after initial treatment. The post-hoc powers were greater than 0.999 for CST, CMV, and PFMV in the comparison between that at baseline and 6 months after initial treatment. We have added this point to the revised manuscript (Page 8, lines 12–14 and Page 12, line 11). Although the number of cases in this study was limited, we were able to confirm that the powers were statistically sufficient. We have also added this sentence to the revised manuscript (Page 14, lines 2–3).

Point 2: Figure 1 – there are too much data the double divided figure is confusing please divide it into 2 separate figures. Additionally, the good quality fundus photography with full ring properties of the MA should be indicated in it.

Response 2: Thank you for your valuable suggestion. We completely agree with it. As per your suggestion, we have divided Figure 1 into two modified figures (Figure 1 and Figure 2) and substituted the case (case 2) for another case (case 9) illustrating the color fundus photography with full ring properties of MA. To avoid confusion among readers, we have revised the OCT slice (from 19/19 slice to 14/19 slice) in case 9 in Figure 3 of the revised manuscript.

Point 3: I am a bit concerned about the statistical presentation. The mean and SD for the 11 patients included? If the normality of the data can be proven (e.g., with Shapiro-Wilk test) this is an option but no information in the statistical analysis mention that part. However, if the normality cannot be proven median and 95%CI is better option possibly with mode – you used the non-parametric test.

Response 3: Thank you for pointing this out. We sincerely apologize for the confusion caused owing to the lack of mention of normality. In this study, we performed the Shapiro–Wilk test for each endpoint at each time point; we have summarized the P-values in the supplementary table below. The Shapiro–Wilk test upon significance demonstrates disruption in the normality. We believe that the validity of assuming normality is maintained in the present results. We, therefore, have summarized the means (standard deviation) and comparisons made using the t-test. In the revised manuscript (Page 6, lines 14–16), we have added the following sentences “The normality was evaluated by performing the Shapiro–Wilk test. After confirming that the data were approximately normally distributed, …”.

Supplementary table for point 3:

The following table represents the results of the Shapiro–Wilk test for each endpoint at each time point with the summarized P-values. All P-values were > 0.05.

Baseline

3 M following initial laser treatment

6 M following initial laser treatment

Changes from 6 M following initial laser minus baseline

BCVA (logMAR)

0.647

0.068

0.093

0.709

CST

0.514

0.813

0.28

0.634

CMV

0.469

0.999

0.279

0.628

PFMV

0.22

0.97

0.996

0.06

M, months; BCVA, best-corrected visual acuity; CST, central subfield thickness; CMV, central macular volume; PFMV, parafoveal macular volume.

Point 4: The authors mention the previous studies in the discussion (line 230 – 236) but no exact comparison can be performed. Please provide the extract of the most relevant study preferably in table or graph for the reader to be able to compare the present study with previous ones and to argumentation of the discussion conducted in the text.

Response 4: Thank you for your valuable comment and suggestion. We completely agree with them. As per your suggestion, we have added a detailed comparison with the most relevant previous studies in the discussion section of the revised manuscript by providing a new table (Table 3) (from Page 12, line 30 to Page 13, line 4).

Point 5: Creation of the figure (graphical equivalent that can be used as graphical abstract) describing the possible mechanisms of creation of MAs and their impact on chronic ME is encouraged, preferably with the changes introduced by laser treatment, but not obligatory. This could greatly improve the visibility of the work.

Response 5: Thank you for your valuable suggestion. We completely agree with it. However, the mechanism of creation of MAs in the chronic phase is still unclear. In this study, we did not assess the findings of OCT angiography and FA, such as the blood flow in MAs and retinal vascular changes before and after direct photocoagulation. We would like to omit the description of the mechanism in this manuscript because it may reduce the scientific validity of this manuscript. We would like to continue research to clarify the mechanism of MA formation in the future.

Point 6: In the one of the last parts of the discussion authors indicate the limitation of their study. Please, provide the defence for each limitation – some are given but not quite sufficient.

Response 6: Thank you for pointing this out. We have added the corresponding statements in the revised manuscript (Page 14, lines 2-3 and Page 14, lines 15-17).

Reviewer 2 Report

Dear authors

Analysis for paper partitions:

1 - Introduction: it is necessary to reform the contents and the drafting of the general part to revise the syntax of the theme

2- Discussion: deepen the discussion on the data on the use and use of drugs in macular edma, missing part in the discussion, max three lines:

PMID: 34830624 ; PMID: 33206392 ; PMID: 28165851 .

3 - Check the bibliographic entries throughout the text, some of which do not conform, review some entries in the references and necessarily insert those referred to in point 2 for the purpose of my acceptance.

4 - Review English grammar and in particular applied scientific English: in particular verb tenses and syntax in the discussion.

Author Response

Response to Reviewer 2 Comments

Point 1: Introduction: it is necessary to reform the contents and the drafting of the general part to revise the syntax of the theme.

Response 1: Thank you for your valuable suggestion. We completely agree with it. We have reformed the contents and drafted the general part in the introduction section of the revised manuscript. Kindly note that the references of the revised manuscript were renumbered to reflect this correction.

Point 2: Discussion: deepen the discussion on the data on the use and use of drugs in macular edma, missing part in the discussion, max three lines: PMID: 34830624 ; PMID: 33206392 ; PMID: 28165851.

Response 2: Thank you for your valuable suggestion. As per your suggestion, based on our data, we have added a discussion of the effect of the drugs administered on the number of MAs in the revised manuscript (Page 13, lines 35–38) following the content of the revised manuscript. We attempted to keep the description as concise as possible.

Point 3: Check the bibliographic entries throughout the text, some of which do not conform, review some entries in the references and necessarily insert those referred to in point 2 for the purpose of my acceptance.

Response 3: Thank you for pointing this out and for your valuable suggestion. As per your suggestion, we checked the bibliographic entries throughout the revised manuscript. Meanwhile, we referred the three articles (PMID: 34830624; PMID: 33206392; PMID: 28165851) you suggested. This manuscript focuses on the efficacy of OCT-guided direct photocoagulation for MAs associated with chronic BRVO, not on the drug therapy for macular edema in diabetic retinopathy or uveitis. We, therefore, consider the inclusion of the aforementioned three articles to be inconsequential to the main purpose of this study and might mislead readers. We sincerely apologize for not adhering to your recommendations. We hope you understand our concern and hesitation.

Point 4: Review English grammar and in particular applied scientific English: in particular verb tenses and syntax in the discussion.

Response 4: Thank you for your valuable comment. We have reviewed the English grammar and applied scientific English in the revised manuscript. The words and syntax of the revised manuscript have also been checked and revised by a native speaker specializing in scientific English.

Round 2

Reviewer 1 Report

You have done a great job in terms of clinically sound problem as well as proper presentation of your work. I am grateful to be able to work as a reviewer for this publication and will look forward for your future work in this subject.